# Outcomes of Patients with Advanced Hepatocellular Carcinoma Receiving Lenvatinib following Immunotherapy: A Real World Evidence Study

**DOI:** 10.3390/cancers15194867

**Published:** 2023-10-06

**Authors:** Mathias E. Palmer, Jennifer J. Gile, Michael H. Storandt, Zhaohui Jin, Tyler J. Zemla, Nguyen H. Tran, Amit Mahipal

**Affiliations:** 1Department of Medicine, Mayo Clinic, Rochester, MN 55905, USA; palmer.mathias@mayo.edu (M.E.P.); storandt.michael@mayo.edu (M.H.S.); 2Department of Oncology, Mayo Clinic, Rochester, MN 55905, USA; gile.jennifer@mayo.edu (J.J.G.); jin.zhaohui@mayo.edu (Z.J.); tran.nguyen@mayo.edu (N.H.T.); 3Department of Clinical Trials and Biostatistics, Mayo Clinic, Rochester, MN 55905, USA; zemla.tyler@mayo.edu; 4Department of Oncology, University Hospitals Seidman Cancer Center, Case Western Reserve University, Cleveland, OH 44106, USA

**Keywords:** hepatocellular cancer, lenvatinib, immunotherapy

## Abstract

**Simple Summary:**

Immunotherapy with atezolizumab plus bevacizumab as well as tremelimumab plus durvalumab has recently become the preferred first-line treatment for advanced hepatocellular carcinoma (HCC). Lenvatinib was initially approved as a first-line treatment, but little is known about its effectiveness following immunotherapy. The aim of our retrospective study was to characterize the clinical outcomes with lenvatinib following immunotherapy treatment. In our cohort of 53 patients, the median progression free survival was 3.7 months, and the median overall survival was 12.8 months from lenvatinib initiation. Multivariate analysis demonstrated race, gender, and Child Pugh Class as significant predictors of survival outcomes and BMI as well as distant metastasis as predictors of progression free survival. This study supports the efficacy of lenvatinib following immunotherapy and validates the use of lenvatinib as a second-line therapy following progression on immunotherapy in patients with advanced HCC.

**Abstract:**

Background: Lenvatinib, a multikinase inhibitor, is an FDA-approved treatment for advanced hepatocellular carcinoma (HCC) in the first-line setting. Recent trial data have established atezolizumab plus bevacizumab as well as tremelimumab plus durvalumab as preferred first-line treatment options for advanced HCC. The role of lenvatinib following progression on immunotherapy in patients with advanced HCC remains unclear. Methods: We conducted a multicentric, retrospective analysis of patients with advanced HCC diagnosed between 2010 and 2021 at the Mayo Clinic in Minnesota, Arizona, and Florida who received immunotherapy followed by lenvatinib. Median overall survival and progression-free survival analyses were performed using the Kaplan–Meier method, and responses were determined using RECIST 1.1. Adverse events were determined using CTCAE v 4.0. Results: We identified 53 patients with advanced HCC who received lenvatinib following progression on immunotherapy. Forty five (85%) patients had a Child Pugh class A at diagnosis, while 30 (58%) patients were still Child Pugh A at time of lenvatinib initiation. Lenvatinib was administered as a second-line treatment in 85% of the patients. The median PFS was 3.7 months (95% CI: 3.2–6.6), and the median OS from the time of lenvatinib initiation was 12.8 months (95% CI: 6.7–19.5). In patients with Child Pugh class A, the median OS and PFS was 14 and 5.2 months, respectively. Race, gender, and Child Pugh class was associated with OS on multivariate analysis. Discussion: Our study, using real-world data, suggests that patients benefit from treatment with lenvatinib following progression on immunotherapy in advanced HCC. The optimal sequencing of therapy for patients with advanced HCC following progression on immunotherapy remains unknown, and these results need to be validated in a clinical trial.

## 1. Introduction

Liver cancer, including hepatocellular carcinoma (HCC), is the seventh most common malignancy and carries the third highest cancer-related mortality worldwide; in the USA there is an incidence of 9.2 per 100,000 [1,2]. HCC is frequently associated with underlying liver disease secondary to infection with Hepatitis B or C, chronic alcohol use, or metabolic-associated steatohepatitis. Unfortunately, liver cancer is commonly diagnosed in the advanced stage where curative therapies are not feasible, with 18% of patients having distant metastatic disease at diagnosis [3].

The treatment of localized HCC includes surgical resection, liver transplant in select patients, ablation, embolization, and radiation therapy. For advanced or metastatic disease, the standard of care treatment typically involves systemic therapy. Patients with HCC usually have a competing mortality risk due to underlying cirrhosis, and the benefit of systemic treatment is limited to patients with preserved liver function. Sorafenib, a small molecule kinase inhibitor of Raf-1, B-Raf, VEGFR1, VEGFR2, VEGFR3, and PDGFR-ß, was the first treatment to demonstrate improved overall survival and progression-free survival. The SHARP trial published in 2008 demonstrated an overall survival (OS) improvement of 10.7 months compared to 7.9 months in the placebo arm with an increased time to radiographic progression of 5.5 months compared to 2.8 months [4].

After more than a decade with no change in the first-line systemic therapy for HCC, lenvatinib, a multikinase inhibitor of FGF1-4, VEGFR1-3, PDGFα, KIT, and RET, was approved for the first-line treatment of unresectable HCC. The FDA approval of lenvatinib was based on the phase 3 REFLECT trial, which randomized 954 patients with advanced HCC and demonstrated noninferiority compared to sorafenib. The median overall survival was 13.6 months on lenvatinib compared to 12.3 months on sorafenib with superiority of secondary endpoints, including progression-free survival, time to progression, and objective response rates [5,6].

While small molecule inhibitors had dominated the field of HCC for many years, a paradigm shift began with the use of immune checkpoint inhibition (ICI) and vascular endothelial growth factor receptor (VEGFR) inhibition. Immune checkpoint inhibitors, such as pembrolizumab and nivolumab, had previously demonstrated robust and durable responses in other solid tumors, such as lung, melanoma, and breast cancer. Initial phase 2 and phase 3 trials, such as CheckMate-040, Keynote-224, and Keynote-240, demonstrated response rates of 7–20% and led to the US Food and Drug Administration (FDA)’s approval as a second-line therapy for advanced HCC [7,8,9].

More recently, IMBRAVE150 and HIMALAYA have demonstrated immune checkpoint inhibitor superiority compared to sorafenib. The phase 3 study IMBRAVE150 compared atezolizumab, a PD-L1 inhibitor, and bevacizumab, a VEGF-A inhibitor, to sorafenib as a first-line therapy and demonstrated their superiority in progression-free survival (PFS) (6.8 months compared to 4.3 months with sorafenib) and OS (19.2 months compared to 13.4 months with sorafenib) (hazard ratio (HR) 0.66, 95% CI 0.52–0.85, *p* < 0.001) [10,11]. HIMALAYA is a phase 3 trial that evaluated tremelimumab, a CTLA-4 inhibitor, and durvalumab, a PD-L1 inhibitor, compared to sorafenib as a first-line therapy for HCC and demonstrated an improved median overall survival (mOS) (16.56 months vs. 13.77 months, HR 0.76, 95% CI 0.65–0.92, *p* = 0.0035) [12]. Since these studies, both atezolizumab plus bevacizumab as well as tremelimumab plus durvalumab have received FDA approval as a first-line treatment of HCC and have replaced tyrosine kinase inhibitors as the standard of care for patients who do not have any contraindication to receive immunotherapy.

However, there is a lack of data evaluating tyrosine kinase inhibitors following the use of immunotherapy. The previous first-line treatment options of lenvatinib and sorafenib are increasingly being used as second-line treatments without any clinical trial data. Because of the changing landscape of HCC treatment, it is unlikely that there will be a trial comparing lenvatinib to other approved agents in a second-line setting following immunotherapy-based treatment for HCC. Real world data could have utility in providing outcomes data that could help guide treatment for this patient population. In this retrospective, single-institution, multicenter study, we aim to characterize outcomes in patients who received lenvatinib following progression on first-line immunotherapy.

## 2. Methods

This is a retrospective review of patients seen within the Mayo Clinic Enterprise, including sites in Minnesota, Arizona, and Florida with a radiologically and/or pathologically confirmed diagnosis of HCC between 1 January 2010 and 31 December 2021 and who had initially received immunotherapy followed by lenvatinib. Patients and their clinical data were obtained via electronic record search using key terms. Immunotherapy treatments included pembrolizumab, nivolumab, atezolizumab plus bevacizumab, and tremelimumab plus durvalumab in combination or as single agent therapy. The assessment of patient characteristics included age at diagnosis, tumor stage and grade at diagnosis, Child Pugh score at diagnosis and at each subsequent line of therapy, age, gender, body mass index (BMI), body surface area (BSA), clinical history, and systemic treatments received. For continuous data, the median and Interquartile Range (IQR) are reported. For categorical data, frequencies and percentages are reported. Institutional review board approval was obtained for this retrospective study, and it was deemed that obtaining inform consent not required.

Primary endpoints included progression-free survival (PFS) during lenvatinib therapy and overall survival (OS) following treatment with lenvatinib in patients who had previous progression on immunotherapy. PFS was defined as time from initiation of lenvatinib until disease progression or death due to any cause. Patients who did not progress or die while on lenvatinib were censored at cessation of lenvatinib therapy. OS was defined as time from initiation of lenvatinib until death from any cause. Patients were censored for OS at the time they were last known to be alive. Both PFS and OS were calculated using Kaplan–Meier method with the median (95% confidence interval, CI) reported. As secondary analyses, all baseline characteristics were univariately evaluated to determine whether they were prognostic predictors of both PFS and OS. A Cox Proportional Hazards (PH) model was utilized with a *p*-value threshold of 0.05 to determine significance. Additionally, a backwards selection Cox PH modeling procedure (threshold *p*-value of 0.05) was utilized to create a multivariable model for both PFS and OS. Testing for proportional hazards was performed for all variables in the final multivariable models for both PFS and OS. It was determined that no assumptions were violated. Response was determined using Response Evaluation Criteria in Solid Tumors (RECIST) version 1.1. Disease control (DC) was defined as having stable disease, partial response, or complete response. The percentage of total patients with DC was calculated. Common Terminology Criteria for Adverse Events (CTCAE) was used for the determination of adverse events.

## 3. Results

A total of 53 patients with HCC were identified who received lenvatinib following immunotherapy. The median age at the initiation of treatment with lenvatinib was 67 years (IQR, 59–72 years), and the majority were male (83%) and of Caucasian descent (75.5%). The median BMI was 26.7 kg/m^2^, and 15 (28.3%) patients had a BMI > 30 mg/m^2^. The median AFP at diagnosis was 33.2 mg/mL (IQR, 5–489 ng/mL). Patient demographics are summarized in Table 1. Thirty patients (57%) underwent liver embolization, and nine (17%) patients underwent prior ablation. At the time of diagnosis 45 (84.9%) patients had Child Pugh class A at diagnosis, and only 30 (56.6%) patients had child Pugh Class A at the time of lenvatinib initiation. In terms of prior immunotherapy treatments, atezolizumab plus bevacizumab was most common treatment (62.3%), followed by nivolumab (22.6%), durvalumab plus tremelimumab (7.5%), and pembrolizumab (7.5%). Lenvatinib was administered as a second-line treatment in 45 (85%) patients.

Survival Outcomes: The median follow up time for the study was 23 (12.9, NE) months. The median PFS on lenvatinib treatment was 3.7 months (95% CI, 3.2–6.6 months) (Figure 1), and the median OS was 12.8 months (95% CI, 6.7–19.5) (Figure 2). The median duration of lenvatinib treatment was 3 months (IQR, 0–29 months). The disease control rate (DCR) was 56.5%, with fourteen patients (30.4%) having partial response (PR), twelve (26.1%) having stable disease (SD), and twenty (43.5%) having progressive disease (PD). A total of 37 patients died during this study.

The Child Pugh score and presence of distant metastasis were significantly associated with both PFS and OS in univariate analysis (Table 2). Other patient characteristics, including age, gender, BMI, AFP, previous embolization, hepatitis status, NASH status, vascular invasion, and number of previous lines, were not significantly associated with PFS nor OS. The median PFS was 5.2 months in patients with Child Pugh class A compared to 3.4 months in those with a Child Pugh score of B7 and 1.9 months in patients with a Child Pugh score of 8 or higher. Similarly, the median OS of 14 months in patients with a Child Pugh score of 5 or 6 was significantly higher than in those with a Child Pugh score of 7 (6.1 months) and 8 or higher (2.4 months). Utilizing a Cox PH model with a backwards selection process, in multivariate analysis, gender (*p* = 0.026), race (*p* = 0.042), and Child Pugh class (*p* = 0.007) were significant predictors of overall survival. BMI (*p* = 0.027) and presence of distant metastases (*p* = 0.013) were significant predictors of progression-free survival in multivariate analysis (Table 3).

The most common grade-3 or -4 side effects included hypertension (38%), fatigue (55%), aspartate transaminase (AST) elevation (32%), anorexia (28%), and nausea (19%). A total of 10 patients discontinued treatment with lenvatinib due to adverse events including hypertension, confusion, fatigue, weakness, elevated bilirubin, Congestive Heart Failure (CHF) exacerbation, anorexia, and arthralgias (Table 4). In this study, we also evaluated the association between the occurrence of grade-3 or -4 adverse events and clinical outcomes. Patients who developed grade-3 or -4 adverse events had significantly prolonged PFS (median PFS: 8.1 vs. 3.4 months; *p* = 0.036) compared to those patients who did not develop severe adverse events. OS was not significantly different in the two groups (median OS: 19.5 vs. 11.1 months; *p* = 0.113).

## 4. Discussion

In this retrospective study, we aimed to describe outcomes of patients with advanced HCC who received lenvatinib following progression on initial immunotherapy. Our data demonstrate patients who received lenvatinib had a median OS of 12.8 months and median progression-free survival (mPFS) of 3.7 months. Importantly, in patients with Child Pugh class A, the median OS and PFS was 14 and 5.2 months, respectively. In addition, 30,4% of the patients achieved partial responses. In the phase 3 REFLECT trial evaluating lenvatinib as a first-line treatment option, a median OS and PFS of 13.6 and 7.4 months, respectively was reported. Our results suggest that lenvatinib retains anti-tumor activity following initial treatment with immunotherapy in the real-world patient population. Further, the benefit of lenvatinib was primarily seen in patients with Child Pugh Class A and, to some extent, in those with a Child Pugh score of 7.

To further characterize the response to treatment with lenvatinib, we performed a multivariable analysis, which demonstrated non-white race, male gender, and Child Pugh class B or C as significant predictors of worse overall survival within our cohort. The differential effect on race and gender remains unclear but could in part be related to etiology of liver disease.

The finding of race as a significant predictor of poor outcomes was surprising and adds to the growing literature of disparity within the healthcare system. This finding, however, must be interpreted cautiously as > 75% of participants were Caucasian, indicating poor power to properly detect outcome differences based on race. On further evaluation of the data, the non-white patients had a relatively short duration of treatment with lenvatinib of just 2.9 months on average compared to the overall mean duration of lenvatinib treatment of 4.8 months. There is currently a paucity of data analyzing racial differences in lenvatinib treatment outcomes; the REFLECT cohort was composed of primarily Asian and white participants, with just 2% of the patients who were non-white and non-Asian. Socioeconomic differences between races may account for some of these differences in outcomes observed in our study, but further research is required to assess socioeconomic impacts on HCC outcomes.

Multivariable analysis demonstrated distant metastasis and BMI as significant predictor of PFS in our cohort. The finding of BMI has not previously been demonstrated but may be related to the association with NASH and increased BMI. The association between distant metastasis and progression-free survival is likely related a combination of distant metastasis being a marker of disease burden at the time of lenvatinib initiation and the increased probability of progression with an increased number of tumor sites. Interestingly, we also report that occurrence of grade-3 or -4 adverse events was associated with improved PFS. The association between adverse events and efficacy has been reported with immunotherapy and other anticancer agents. However, this finding needs to be validated in a larger study.

One retrospective study by Qin et al. [13] evaluated lenvatinib as a second-line therapy in 50 patients with HCC. In this study, 60% previously received sorafenib, and 40% received immune checkpoint inhibitors as a first-line therapy; 46% of patients were child Pugh B at inclusion. They reported an mOS of 8.5 months and an mPFS of 5 months with a disease control rate of 74% and noted that DCR was improved in patients who received immunotherapy as a first-line therapy compared to sorafenib as a first-line therapy. The Child Pugh score was a significant predictor of response to lenvatinib, similar to our study.

To our knowledge this is the largest study evaluating the role of lenvatinib following immunotherapy in patients with advanced HCC. In addition, this study addresses the paucity of data available in the Western population. The shortcomings of this study includes the retrospective nature, which could have confounding and selection bias. There was likely a selection bias of patients healthy enough to receive lenvatinib following treatment with immunotherapy, which would be true for any second-line therapy trial. We did include the multivariate analysis to partly address this issue. The study was conducted at a tertiary care center, which could limit the generalizability of the results. The study did include patients from varied geographical area, including Florida, Arizona, and Minnesota. There was heterogeneity in immunotherapy timing and treatment choice as the studies demonstrated that the efficacy of neither atezolizumab plus bevacizumab nor tremelimumab plus durvalumab had been published at the time of immunotherapy treatment in this study. We believe that in future studies, having a standard immunotherapy regimen would provide valuable insights. Finally, despite the fact that this is one of the largest studies in the Western population, the relatively small sample size of this study would require validation through a larger cohort to confirm these results. Also, different immunotherapy agents were utilized prior to initiating therapy with atezolizumab plus bevacizumab, which is the most common regimen. However, in our study, the choice of prior immunotherapy agent did not seem to make a difference in efficacy outcomes.

As a selective inhibitor of VEGFR1-3, FGFR1-4, PDGFR, KIT, and αRET, lenvatinib targets a combination of alternative and common pathways with the immunotherapy treatment. Immune therapy resistance is hypothesized to be related to the modulation of the immune microenvironment, including the tumor mutation burden, MLH expression, and changes in local cytokines [14]. Lenvatinib acts independently of these tumor microenvironment changes which may be responsible for the preserved efficacy after progression on immunotherapy.

In summary, this study suggests that lenvatinib remains efficacious as a second-line treatment for HCC following progression on immunotherapy. Lenvatinib was generally well tolerated, with 21 (39.6%) of patients experiencing grade-3 or -4 adverse events and 10 (19%) patients discontinuing therapy due to side effects. The side effect profile and rates were similar to those observed in the REFLECT study [6].

As atezolizumab–bevacizumab is now the standard first-line therapy for advanced HCC, it will be essential for additional studies to evaluate the optimal choice and sequence of subsequent lines of therapy. Unless a clinical trial is conducted, the data for a second-line choice would be determined by provider and patient preference, co-morbidities, and data from real-world settings. Our study suggests that lenvatinib could be one viable option. In addition, evaluating the role of lenvatinib in combination with additional therapies may provide additional benefit. The LAUNCH trial demonstrated improved outcomes when lenvatinib was combined with transarterial chemoembolization in a first-line setting and may warrant utility in a second-line setting [15]. In contrast, in the LEAP002 trial, pembrolizumab in combination with lenvatinib did not confer additional benefits in HCC, although this could partly be due to longer median OS with lenvatinib alone in the control arm [16,17]. Additional ongoing clinical trials are evaluating the role of lenvatinib in combination with immunotherapy in both first- and second-line settings (NCT04368078, NCT04770896).

## 5. Conclusions

This study served to characterize the outcomes of patients receiving therapy with lenvatinib following progression on immunotherapy. Patients who received lenvatinib had a median OS of 12.1 months and a median PFS of 3.7 months, which suggests continued efficacy of lenvatinib following immunotherapy. In particular, in patients with Child Pugh class A, the median OS of 14 months is similar to that reported with lenvatinib in a first-line setting. Significant predictors of OS included race, gender, and Child-Pugh Score. Additional studies are needed to further delineate the optimal sequence of treatment following progression on immunotherapy and validate our findings.

## Figures and Tables

**Figure 1 cancers-15-04867-f001:**
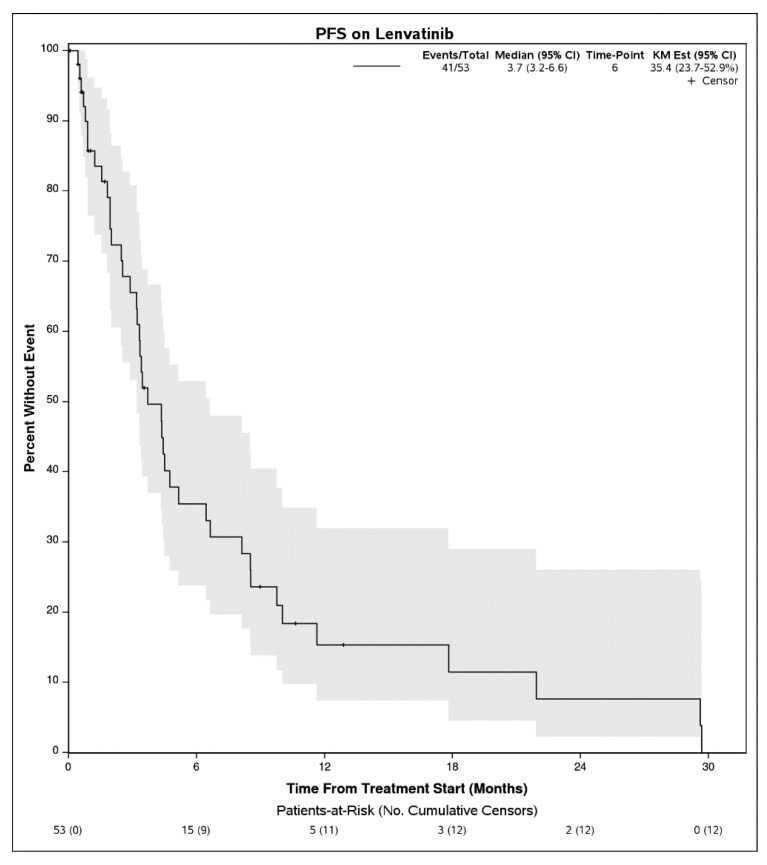
Progression-free survival for patients with advanced hepatocellular carcinoma who received lenvatinib following progression on immunotherapy.

**Figure 2 cancers-15-04867-f002:**
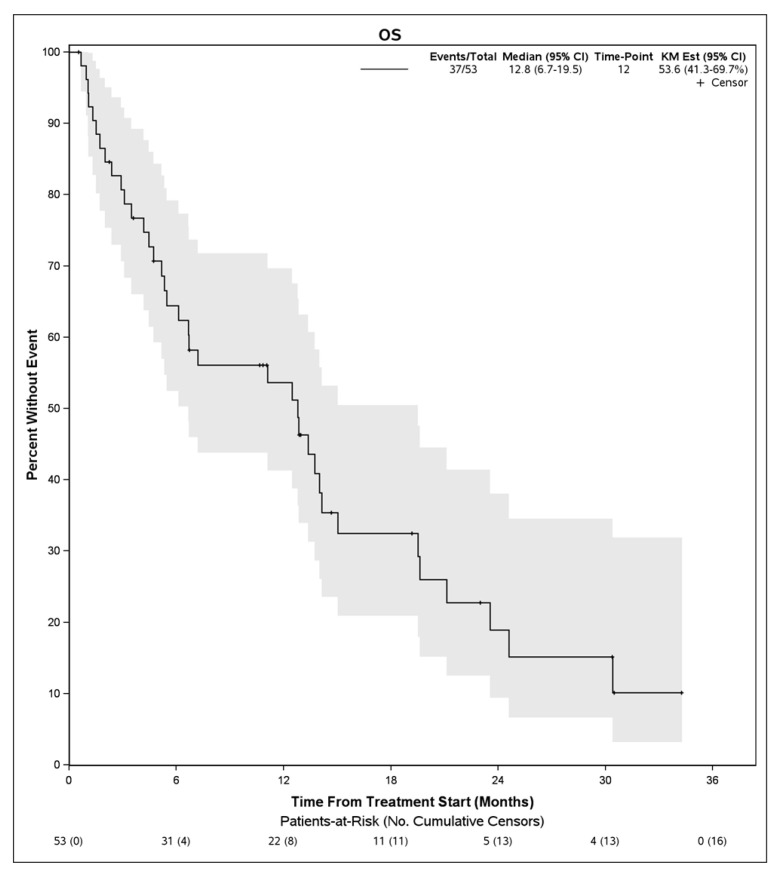
Overall survival from initiation of lenvatinib in patients with advanced hepatocellular carcinoma who progressed initially on immunotherapy.

**Table 1 cancers-15-04867-t001:** Baseline characteristics.

Characteristic	Total (N = 53)
Age at Treatment, years	
Mean (SD)	65.4 (11.5)
Median	67.0
Q1, Q3	59.0, 72.0
Range	(39.0–84.0)
Gender	
Male	44 (83.0%)
Female	9 (17.0%)
Race	
White	40 (75.5%)
Asian	3 (5.7%)
African American	4 (7.5%)
Unknown	4 (7.5%)
Hispanic	2 (3.8%)
BMI	
Median	26.7
Q1, Q3	23.8, 30.7
Range	(19.7–47.9)
AFP at diagnosis	
Median	33.2
Q1, Q3	5.0, 489.0
Range	(1.3–122335.0)
Prior embolization	
No	23 (43.4%)
Yes	30 (56.6%)
Prior ablation	
No	44 (83.0%)
Yes	9 (17.0%)
Prior radiation therapy	
No	47 (88.7%)
Yes	6 (11.3%)
Diabetes mellitus	
No	33 (62.3%)
Yes	20 (37.7%)
Hyperlipidemia	
No	32 (60.4%)
Yes	21 (39.6%)
Alcohol abuse	
No	35 (66.0%)
Yes	18 (34.0%)
History of hepatitis B	
No	49 (92.5%)
Yes	4 (7.5%)
History of hepatitis C	
No	30 (56.6%)
Yes	23 (43.4%)
Metabolic associated steatohepatitis	
No	45 (84.9%)
Yes	8 (15.1%)
Child Pugh score at diagnosis	
Missing	4
A5	32 (65.3%)
A6	13 (26.5%)
B7	4 (8.2%)
Child Pugh score at start of lenvatinib	
Missing	2
A5	23 (45.1%)
A6	7 (13.7%)
B7	11 (21.6%)
B8	4 (7.8%)
B9	4 (7.8%)
C10	1 (2.0%)
C11	1 (2.0%)
Prior immunotherapy	
Atezolizumab + Bevacizumab	33 (62.3%)
Nivolumab	12 (22.6%)
Pembrolizumab	4 (7.5%)
Durvalumab + Tremelimumab	4 (7.5%)
Number of prior lines of treatment	
1	45 (84.9%)
2	6 (11.3%)
3	1 (1.9%)
4	1 (1.9%)
Vascular invasion	
No	43 (81.1%)
Yes	10 (18.9%)
Distant metastasis	
No	30 (56.6%)
Yes	23 (43.4%)
Grade-3/Grade-4 adverse events	
No	32 (60.4%)
Yes	21 (39.6%)
Duration of lenvatinib treatment (months)	
N	53
Mean (SD)	5.4 (6.6)
Median	3.3
Q1, Q3	1.0, 6.6
Range	(0.1–29.7)

**Table 2 cancers-15-04867-t002:** Univariate analysis of progression-free survival (PFS) and overall survival (OS) among patients treated with lenvatinib.

	Overall Survival	Progression-Free Survival		
Variable	HR (95% CI)	*p*-Value	HR (95% CI)	*p*-Value
Age (Continuous)	1.010 (0.865, 1.179)	0.899	0.935 (0.813, 1.075)	0.343
Age (≥65 years)	1.050 (0.532, 2.073)	0.889	0.528 (0.264, 1.057)	0.071
Gender (Male)	2.560 (0.903, 7.260)	0.077	1.567 (0.686, 3.581)	0.287
BMI (Continuous)	0.874 (0.641, 1.193)	0.397	0.744 (0.543, 1.019)	0.065
BMI (≥30)	0.860 (0.419, 1.763)	0.680	0.583 (0.273, 1.244)	0.163
Race (Non-white)	1.203 (0.563, 2.568)	0.634	1.794 (0.881, 3.655)	0.108
AFP (≥5)	0.863 (0.398, 1.869)	0.708	0.871 (0.404, 1.879)	0.725
Prior Embolization (Yes)	1.096 (0.566, 2.123)	0.786	1.463 (0.773, 2.769)	0.243
Hep B (Positive)	1.593 (0.477, 5.322)	0.449	1.435 (0.506, 4.073)	0.497
Hep C (Positive)	0.960 (0.496, 1.857)	0.903	1.181 (0.619, 2.254)	0.614
Hep B/C Status (B or C Positive)	1.076 (0.562, 2.060)	0.825	1.322 (0.697, 2.506)	0.393
NASH (Positive)	1.339 (0.519, 3.458)	0.546	0.655 (0.228, 1.879)	0.431
Child Pugh (B8–C11 vs. A5–6)	5.813 (2.309, 14.633)	0.000	3.636 (1.417, 9.328)	0.007
Child Pugh (B7 vs. A5–6)	2.511 (1.109, 5.684)	0.027	1.595 (0.702, 3.625)	0.265
Number of Prior Lines (≥2)	0.889 (0.311, 2.544)	0.827	1.112 (0.431, 2.870)	0.826
Vascular Invasion (Positive)	2.049 (0.975, 4.306)	0.058	1.118 (0.541, 2.610)	0.667
Distant Metastasis (Positive)	2.148 (1.110, 4.156)	0.023	2.436 (1.239, 4.793)	0.010

**Table 3 cancers-15-04867-t003:** Multivariate Analysis of progression-free survival and overall survival in patients treated with lenvatinib.

**Overall Survival**			
**Variable**	**Comparison**	**HR (95% CI)**	** *p* ** **-Value**
Gender	Male vs. Female	4.578 (1.197, 17.515)	0.026
Race	Other vs. White	2.895 (1.041, 8.047)	0.042
Child Pugh			0.007 *
Child Pugh	B7 vs. A5/A6	2.051 (0.867, 4.851)	0.102
Child Pugh	B8/B9/C10/C11 vs. A5/A6	4.851 (1.792, 13.135)	0.002
**Progression Free Survival**			
Variable	**Comparison**	**HR (95% CI)**	** *p* ** **-Value**
BMI	≥30 vs. < 30	0.386 (0.166, 0.898)	0.0271
Distant Metastases	Yes vs. No	2.701 (1.229, 5.935)	0.0134

* Overall *p*-Value.

**Table 4 cancers-15-04867-t004:** Adverse events experienced on lenvatinib therapy.

	Grade 1 or 2, N (%)	Grade 3 or 4, N (%)
Any	44 (83)	21 (39.6)
Fatigue	28 (52.8)	1 (1.8)
AST Elevation	16 (30.1)	1 (1.8)
Anorexia	14 (26.4)	1 (1.8)
Diarrhea	12 (22.6)	1 (1.8)
Elevated Bili	12 (22.6)	1 (1.8)
Nausea	10 (18.8)	0 (0)
Abdominal Pain	10 (18.8)	0 (0)
Alkaline Phos	10 (18.8)	0 (0)
Hypothyroidism	9 (16.9)	0 (0)
Anemia	9 (16.9)	0 (0)
Limb Edema	8 (15)	0 (0)
Thrombocytopenia	8 (15)	0 (0)
Hypertension	7 (13.2)	13 (24.5)
Pruritis	7 (13.2)	0 (0)
Myalgia	6 (11.3)	0 (0)
Weight Loss	5 (9.4)	1 (1.8)
Palmar plantar erythrodysesthesia	5 (9.4)	1 (1.8)
ALT ↑	5 (9.4)	1 (1.8)
Mucositis	4 (7.5)	1 (1.8)
Abdominal distention	3 (5.6)	0 (0)
Ascites	3 (5.6)	0 (0)
Hoarseness	3 (5.6)	1 (1.8)
Shortness of Breath	3 (5.6)	0 (0)
Hyponatremia	3 (5.6)	0 (0)
Confusion	2 (3.7)	4 (7.5)
Headache	2 (3.7)	0 (0)
Weakness	2 (3.7)	1 (1.8)
Hypoalbuminemia	2 (3.7)	0 (0)
Arthralgia	2 (3.7)	1 (1.8)
Dry Skin	1 (1.8)	0 (0)
Depression	1 (1.8)	0 (0)
GI Bleed	1 (1.8)	0 (0)
Pancreatitis	1 (1.8)	0 (0)
Colitis	1 (1.8)	0 (0)
Back Pain	1 (1.8)	0 (0)
Dizziness	1 (1.8)	0 (0)
Alopecia	1 (1.8)	0 (0)
Dry Eyes	1 (1.8)	0 (0)
Proteinuria	1 (1.8)	1 (1.8)
Epistaxis	1 (1.8)	0 (0)
Gait Disturbance	1 (1.8)	0 (0)
Dysphagia	1 (1.8)	0 (0)
Bradycardia	1 (1.8)	0 (0)
Hyperphosphatemia	1 (1.8)	0 (0)
Lymphopenia	1 (1.8)	0 (0)
Hemoptysis	1 (1.8)	0 (0)
Elevated Creatinine	1 (1.8)	0 (0)
Constipation	1 (1.8)	0 (0)
0	1 (1.8)	0 (0)
CHF	0 (0)	1 (1.8)
Hyperkalemia	0 (0)	1 (1.8)

## Data Availability

The data presented in this study is available upon request from the corresponding author. Due to institutional restrictions, the data are not publicly available.

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
