# Peer review of "Outcomes of Patients with Advanced Hepatocellular Carcinoma Receiving Lenvatinib following Immunotherapy: A Real World Evidence Study"

_cancers, 2023, doi:10.3390/cancers15194867_

Round 1

Reviewer 1 Report

From a biostats and clinical epidemiology point of view, this RWE research has been well planned, here are some suggestions for the Authors:

- title, adding “a real world evidence study” will be of help

- line 23 “following progression on immunotherapy” do you mean either atezolizumab plus bevacizumab or tremelimumab plus durvalumab, only these 2 regimens!? At line 98, you cited nivolumab too, in table 2 tremelimumab is absent: thus, this point deserves to be clarified

- line 29 “studu” typo

- line 102 “For continuous data, the median and range are reported” median/IQR would be better

- figure 1 “Lenvima” use chemical names, not the branded one

- line 132, previous CT/IT lines shows a large heterogeneity, this point should be underlined

- line 136 “median follow up time for the study was 35 months” estimated for alive pts only!?

- results, report all p-values with 3-sign digits

- results, what about any potential PH assumption violations per time-varying covariates!?

- results, OS and PFS crude number of events is lacking and has to be added, even to check the above cited point: this is a critical topic

- table 3, overall p-value has to be reported in an upper, dedicated row. The current position would counfound the reader

- table 3, here in Europe nobody would accept the word “race” (WWII...), preferring “ethnicity”. Mind the clear interaction that this determinant show in the multivariable model, have you tested it?

- table 4, extremely informative!

- line 173 “mPFS” obvious, but undefined

- discussion, the differences in OS vs PFS modeling results should be cited and properly commented

Author Response

From a biostats and clinical epidemiology point of view, this RWE research has been well planned, here are some suggestions for the Authors:

  1. title, adding “a real world evidence study” will be of help

Response: This is an excellent suggestion and we have edited the title.

  1. line 23 “following progression on immunotherapy” do you mean either atezolizumab plus bevacizumab or tremelimumab plus durvalumab, only these 2 regimens!? At line 98, you cited nivolumab too, in table 2 tremelimumab is absent: thus, this point deserves to be clarified

Response: Due to the timing of this study in relation to the clinical trials of immunotherapy in HCC and subsequent FDA approvals, we included any immunotherapy regimens including single therapy and combination therapy. The breakdown of specific immunotherapytherapy regimens has been clarified within the manuscript.

  1. line 29 “studu” typo

Response: We have corrected the error.

  1. line 102 “For continuous data, the median and range are reported” median/IQR would be better

Response: We have updated the manuscript to relfect the IQR rather than the range.

  1. figure 1 “Lenvima” use chemical names, not the branded one

Response: We have corrected the figure.

  1. line 132, previous CT/IT lines shows a large heterogeneity, this point should be underlined

Response: Thanks for mentioning this. We have added the comment about heterogeneity of regimens in the discussion section. Added to the discussion: “There was heterogeneity in immunotherapy timing and treatment choice as the studies demonstrating efficacy of atezolizumab plus bevacizumab nor tremelimumab plus durvalumab had not yet been published at the time of immunotherapy treatment in this study. We believe that in future studies, having a standard immunotherapy regi-men would provide valuable insights.”

  1. line 136 “median follow up time for the study was 35 months” estimated for alive pts only!?

Response: Our initial calculation was based on the time from diagnosis; we have recalculated based on the time of lenvatinib initiation. We used the reverse Kaplan-meir method. Manuscript has been updated to reflect the correct numbers.

  1. results, report all p-values with 3-sign digits

Response: p-values have been adjusted to 3 decimal places

  1. results, what about any potential PH assumption violations per time-varying covariates!?

Response: Testing for proportional hazards was done for all variables in the final multivar-iable models for both PFS and OS. It was determined that no assumptions were violated. Manuscript has been updated including this statement.

  1. results, OS and PFS crude number of events is lacking and has to be added, even to check the above cited point: this is a critical topic

Response: Tahnks for the suggestion. Results section amended to include the number of deaths during this study. The figures include the number of events as well.

  1. table 3, overall p-value has to be reported in an upper, dedicated row. The current position would confound the reader

Response: Table has been updated as suggested

  1. table 3, here in Europe nobody would accept the word “race” (WWII...), preferring “ethnicity”. Mind the clear interaction that this determinant show in the multivariable model, have you tested it?

Response: We did a literature search regarding race vs ethnicity. Race is identifying a group based on their physical traits where as ethnicity is related to cultural expression and identification. The data collected during this study was based on physical characteristics and would be most consistent with race rather than ethnicity. We do not have ethnicity data available for this study. Race is self reported by patients and is part of standard medical records in United States.

  1. table 4, extremely informative!

Response: Thanks for the comment

  1. line 173 “mPFS” obvious, but undefined

Response: We have defined this acronym.

  1. discussion, the differences in OS vs PFS modeling results should be cited and properly commented

Response: Thanks for the suggestion. Based on other reviewers comments, we repeated the modeling after including 2 new variables: vascular invasion and presence of distant metastases. The new modeling results are now included.

Reviewer 2 Report

Interesting study. Major limitations are the limited sample size and the retrospective design, and these limitations should be addressed as such in the Discussion.

Please avoid the use of brand name ("Lenvima")

THe authors should focus on the adverse events that patients experienced under immunotherapy and the general safety profile of these drugs, with particular reference to the risk of re-activation of pre-existing immuno-diseases (cite the recent MA: PMID: 33314269)

Author Response

Interesting study. Major limitations are the limited sample size and the retrospective design, and these limitations should be addressed as such in the Discussion.

Response: We agree with the reviewer that limited sample design and retrospective nature of the study is the limitation for this study. These limitations are included in the discussion section.

  1. Please avoid the use of brand name ("Lenvima")

Respoonse: Thanks for the suggestion. We have corrected this figure and have utilized the generic name of lenvanib.

  1. The authors should focus on the adverse events that patients experienced under immunotherapy and the general safety profile of these drugs, with particular reference to the risk of re-activation of pre-existing immuno-diseases (cite the recent MA: PMID: 33314269)
    1. Add the above statement on immunotherapy safety to the discussion.
    2. The data collected during this study occurred after patinets discontinued immunotherapy and began treatment with lenvatinib and was collected in a retrospective nature by reviewing clinical documentation with providers while on lenvatinib. There were no explicit cases mentioned of side effects directly attributed to the immunotherpy treatment. As a result, side effects collected during this study cannot be strictly attributed to the initation of immunotherpay or the relation to reactivation of pre-existing immuno-diseases.

Response: In this study, we looked at the outcome of patients who received lenvatinib after progression on immunotherapy agents. The adverse events that we report are related to lenvatinib treatment. We did not collect adverse events on prior immunotherapy treatment. We did not observe any reactivation of pre-existing autoimmne disease with lenvatinib.

Reviewer 3 Report

This paper demonstrates the efficacy of lenvatinib in second-line and beyond therapy for unresectable hepatocellular carcinoma; the evidence for second-line systemic therapy is insufficient, and we believe this information will be helpful in real-world practice.

Please consider the following points.

Major

1) The author used "backward selection" to analyze factors in the multivariate analysis; however, please reconsider whether this method is appropriate. Because it is hard to understand that racial differences that were not significant in the univariate analysis became a significant factor in the multivariate analysis. The reviewer thinks race might be confounding for other factors, such as tumor factors. 

2) The reviewer thinks there is a lack of information on tumor factors, including the presence or absence of distant metastasis, vascular invasion, and BCLC stage in the patient background. These factors must also be analyzed as factors related to OS and PFS.

3) The rationale for setting the AFP cutoff at 5 for factors related to OS and PFS is needed.

Minor

1) The authors described the median observation period as 35 months, while the median OS of this cohort was 12.8 months, which is inconsistent. Does this mean the starting point for the observation period and OS are different?

2) I think one of "received" and "underwent" on Page 3 and line 129 should be deleted.

3) In Table 2, "sex" is used, and in Table 3, "gender" is used, and I think it is desirable to unify either of them.

Author Response

This paper demonstrates the efficacy of lenvatinib in second-line and beyond therapy for unresectable hepatocellular carcinoma; the evidence for second-line systemic therapy is insufficient, and we believe this information will be helpful in real-world practice.

Please consider the following points.

Major

  1. The author used "backward selection" to analyze factors in the multivariate analysis; however, please reconsider whether this method is appropriate. Because it is hard to understand that racial differences that were not significant in the univariate analysis became a significant factor in the multivariate analysis. The reviewer thinks race might be confounding for other factors, such as tumor factors. 

Response: Thank you for raising this question. We have re-evaluated the model to include tumor factors of distant metastasis and vascular invasion. These results have been now incorporated into our manuscript. The addition of these two variables did change the results of the multivariate analysis especially with reference to progression-free survival.

  1. The reviewer thinks there is a lack of information on tumor factors, including the presence or absence of distant metastasis, vascular invasion, and BCLC stage in the patient background. These factors must also be analyzed as factors related to OS and PFS.

Response: Thanks for the suggestion. We have now included vascular invation and distant metastasis into the analaysis. These patients were all BCLC C. We redid the analysis with the additional factors and updated results are included in the manuscript.

  1. The rationale for setting the AFP cutoff at 5 for factors related to OS and PFS is needed.

Response: At the time of the study, at our lab, AFP of 5 was a cutoff for normal range.

Minor

  1. The authors described the median observation period as 35 months, while the median OS of this cohort was 12.8 months, which is inconsistent. Does this mean the starting point for the observation period and OS are different? 

Response: We initially we calculated the median observation period based on the time of diagnosis; we have adjusted to include the median time from initiation of lenvatinib. This was calculated using reverse Kaplan-Meir method

  1. I think one of "received" and "underwent" on Page 3 and line 129 should be deleted. 

Response: We have edited as suggested.

  1. In Table 2, "sex" is used, and in Table 3, "gender" is used, and I think it is desirable to unify either of them. 

Response: We agree with the reviewer. We have edited the manuscript to include “gender” consistently.

Reviewer 4 Report

The authors performed analysis of existing HCC patient data to form and test their hypothesis that treatment with Lenvatinib after immunotherapy can be beneficial in advanced cases of HCC. The article is well written and follows systematic structure to present the findings. However, some language related errors persist, that need to be addressed. I also have some suggestions that may help improve the manuscript. 

1.      Following lines: Line 26, line 47, line 98, line 175, line 184 need to be corrected for language/writing errors.

2.      The authors ran analysis on several patient characteristics to test statistical correlation with PFS and OS. However, in the results section they only discuss Child Pugh score, other characteristics although not significant needs to be discussed.

3.      If the authors can provide an expert commentary/rationale on why this treatment strategy may work specifically in advanced HCC, that would be a great addition to the discussion section.

4.      The authors provide a list of adverse events that were observed post-treatment. Is there a correlation between high grade adverse events and better PFS/OS. Are the patients with improved PFS/OS are the ones that have higher grade adverse events or the opposite is true? This analysis could also be a useful addition to the manuscript and could help in formulating future strategy.

Listed in the comments and suggestions for authors section

Author Response

The authors performed analysis of existing HCC patient data to form and test their hypothesis that treatment with Lenvatinib after immunotherapy can be beneficial in advanced cases of HCC. The article is well written and follows systematic structure to present the findings. However, some language related errors persist, that need to be addressed. I also have some suggestions that may help improve the manuscript. 

  1. Following lines: Line 26, line 47, line 98, line 175, line 184 need to be corrected for language/writing errors.

Response: We have corrected the errors as suggested.

  1. The authors ran analysis on several patient characteristics to test statistical correlation with PFS and OS. However, in the results section they only discuss Child Pugh score, other characteristics although not significant needs to be discussed.

Response: We included 2 additional factors as suggested by other reviewer (vascular invasion and presence of distant metastases). We did rerun the analysis and have updated the results. We have included non significannt factors in the results section as well.  

  1. If the authors can provide an expert commentary/rationale on why this treatment strategy may work specifically in advanced HCC, that would be a great addition to the discussion section.

Response: We have added additional commentary including possible mechanisms of action for lenvatinib following immunotherapy in the discussion section.

  1. The authors provide a list of adverse events that were observed post-treatment. Is there a correlation between high grade adverse events and better PFS/OS. Are the patients with improved PFS/OS are the ones that have higher grade adverse events or the opposite is true? This analysis could also be a useful addition to the manuscript and could help in formulating future strategy.

Response: Thanks for the suggestion. As suggested, we evaluated if grade 3/4AE was associated with PFS and OS. Occurrence of grade 3 or 4 AE was associated with improved PFS but not OS. This has been included in the manuscript.

Round 2

Reviewer 1 Report

The Authors were able to solve all previous concerns

Reviewer 2 Report

The revised version of the paper is OK

Reviewer 3 Report

The author responded the reviewer's comments appropriately.